# Efficient mRNA Delivery with Lyophilized Human Serum Albumin-Based Nanobubbles

**DOI:** 10.3390/nano13071283

**Published:** 2023-04-05

**Authors:** Hiroshi Kida, Yutaro Yamasaki, Loreto B. Feril Jr., Hitomi Endo, Keiji Itaka, Katsuro Tachibana

**Affiliations:** 1Department of Anatomy, Faculty of Medicine, Fukuoka University, 7-45-1 Nanakuma, Jonan-ku, Fukuoka 814-0180, Japan; 2Department of Biofunction Research, Institute of Biomaterials and Bioengineering, Tokyo Medical and Dental University (TMDU), 2-3-10 Kanda-Surugadai, Tokyo 101-0062, Japan

**Keywords:** nanobubble, lyophilization, sonoporation, mRNA, drug delivery system, ultrasound

## Abstract

In this study, we developed an efficient mRNA delivery vehicle by optimizing a lyophilization method for preserving human serum albumin-based nanobubbles (HSA-NBs), bypassing the need for artificial stabilizers. The morphology of the lyophilized material was verified using scanning electron microscopy, and the concentration, size, and mass of regenerated HSA-NBs were verified using flow cytometry, nanoparticle tracking analysis, and resonance mass measurements, and compared to those before lyophilization. The study also evaluated the response of HSA-NBs to 1 MHz ultrasound irradiation and their ultrasound (US) contrast effect. The functionality of the regenerated HSA-NBs was confirmed by an increased expression of intracellularly transferred Gluc mRNA, with increasing intensity of US irradiation. The results indicated that HSA-NBs retained their structural and functional integrity markedly, post-lyophilization. These findings support the potential of lyophilized HSA-NBs, as efficient imaging, and drug delivery systems for various medical applications.

## 1. Introduction

Over the past half century, the combination of ultrasound (US) and bubble reagents for medical applications has been researched and significantly evolved. The most well-studied applications of bubble reagents are US contrast agents, drug delivery enhancers, and their combined “theranostics” systems [1].

Bubble reagents function as a US contrast agent. Ultrasounds propagate through biological tissue as a medium, but when bubbles exist in water-filled blood vessels, the acoustic impedance changes at the gas-liquid interface and is captured as a reflected echo [2]. Strong echogenicity is observed in bubbles with flexible surfaces when US is applied [3]. Microbubbles (MBs) can be used as US contrast agents based on this principle, thus, US contrast agents tend to be stabilized with insoluble gases and shell materials [4].

These advances in US contrast agents have provided a foundation for US-mediated drug delivery systems (DDS). Bubbles in water pulsate at their natural frequency in resonance with the frequency of sound according to a phenomenon known as Minnaert resonance [5] derived from the Rayleigh-Plesset equation [6]. When irradiated with US, the low- and high-pressure states are repeated in a frequency-synchronized manner within the sound field. Bubbles intermittently oscillate (expand and contract repeatedly), grow while entering the surrounding gas molecules, and eventually burst and collapse. This growth and collapse phenomenon of bubbles under an acoustic field is known as “acoustic cavitation” [7]. The microjet flow, created by the collapse of bubbles, can perforate a cell membrane and allow drugs or genes to penetrate the cytoplasm. In the 1990s, it was reported that US contrast agent MBs acted as nuclei to lower the acoustic cavitation threshold [8]. Tachibana et al. first reported in 1995 that a micro-liquid jet flow generated during the disintegration of human serum albumin (HSA)-based MBs can enhance drug efficacy [9]. In their report, the US irradiation of thrombus in vitro in the presence of US contrast agents, thrombolytic agent, and urokinase was shown to enhance thrombolysis. Several DDS studies have since been conducted using combinations of US and MBs. High-velocity microjets induced during ultrasonically irradiated bubble collapse can also be used to transiently perforate cell membranes [10]. This phenomenon allows extracellular drugs and genes to penetrate living cells [11,12,13,14]. Unger et al. first named their DDS technique “sonoporation,” now a widely used term, which utilizes the acoustic cavitation of MBs [15,16]. Alternatively, in non-inertial cavitation, the bubbles steadily oscillate. The repeated contraction and expansion of bubbles cleave tight junctions in the vessel wall, such as the blood–brain barrier (BBB), allowing drugs to migrate from the vascular lumen into the parenchyma of the organ. A drug delivery system called “BBB opening,” which takes advantage of this phenomenon, is currently under development [17].

MBs of 1–10 µm in diameter have been used in previous studies [9]. However, in recent years, nanobubbles (NBs), which are officially termed as ultra-fine bubbles [18] and are sub-micron diameter bubbles, smaller than MBs, are expected to be a more promising biomaterial owing to their unique properties [19,20]. First, the flotation velocity of an NB, which has a much smaller grain size than an MB, is negligibly small in calculations based on Stokes’ equation [21]. It has also been reported that NBs are stable in water for several months, although they are impaired by freeze-thaw cycles [22]. Lastly, NBs can penetrate deeper into living tissues in vivo, than MBs. One of the characteristic actions of nano-sized biomaterials is the enhanced permeability and retention (EPR) effect [23,24]. In solid tumors, a wide capillary endothelial cell gap exists that is not present in normal tissue. High-molecular-weight diagnostic and therapeutic agents of several hundred nanometers can pass through and accumulate specifically in tumor tissues without being excreted. This effect, called the EPR effect, is important for drug delivery in cancer treatment. EPR effect is expected to occur only in NBs that are particularly small, specifically, less than 200 nm in diameter [25,26,27].

One method of improving the shelf-life of drugs, including bubble reagent stability by shells, is lyophilization. These reagents are generated by evaporating only the liquid, thus preserving the shell morphology. To date, previous reports have shown that MBs and NBs with shells composed of phospholipids or polymers can be successfully preserved by lyophilization [28,29,30,31,32,33,34]. However, owing to technical difficulties only one previous report on the lyophilization of MBs shelled with HSA has been reported [35]. In this study, we demonstrate the successful lyophilization of NBs with HSA shells. This method preserved NB morphology, conservation, and regeneration efficiency, and, ultimately, the functionality of regenerated NBs as a contrast agent and cavitation nucleus is demonstrated in this study.

## 2. Materials and Methods

### 2.1. Preparation of NB Solutions and Their Lyophilized Materials

Human serum albumin-based NBs (HSA-NBs) were prepared using a modified version of a previous study [36]. Briefly, 5 mL of perfluoropropane gas (C3F8; Takachiho Chemical Industrial, Tokyo, Japan) was injected into a free, empty 5 mL glass vial. The vial, which was sealed with a rubber stopper and an aluminum cap, was filled with an additional 1.5 mL of perfluoropropane gas and 3.6 mL of a distilled water solution of 0.06% HSA (Albuminar-25; CSL Behring LLC, IL, USA) using a 23-gauge needle inserted through the rubber stopper. The vial was placed in a high-speed shaking-type tissue homogenizer (Precellys Evolution; Bertin Instruments, FR) and shaken at 6000 rpm for 80 s. After shaking, the vial was centrifuged at 100× *g* for 2 min (MX-301; TOMY, Tokyo, Japan) and cooled on ice for 3 min. The course of shaking, centrifugation, and cooling was repeated thrice. The shaking phase was conducted once more. An 18-gauge needle was inserted into the rubber stopper of the vial to reduce pressure. Temperature changes in the HSA solution due to vibrations and cooling were measured over time using a digital thermometer (TX1003; Yokogawa, Tokyo, Japan).

Solutions containing MBs and NBs in vials perforated with rubber stoppers were immersed in liquid nitrogen and flash-frozen. The frozen material was lyophilized using a freeze dryer (FreeZone 4.5; LABCONCO, MO, USA) for 48 h at a vacuum pressure under 20 Pa and at a collector temperature of -50° C. The vials containing lyophilized NBs were again rubber stoppered and aluminum sealed to prevent wetting and stored at 4 °C.

If not lyophilized, the suspension containing bubbles was mixed uniformly by pipetting and stored at 4 °C until measurements were performed within 2 h.

### 2.2. Morphological Observation of NB Lyophilized Material with Scanning Electron Microscopy

The lyophilized material of the NBs and the control was carefully torn off using tweezers and mounted on aluminum stubs with carbon tape with the cross-section facing upward. The material cross-section was coated with an osmium layer 30 nm in thickness using an OPC-80 osmium plasma coater (Nippon Laser & Electronics Lab. Nagoya, Japan). The samples were observed using scanning electron microscopy (SEM) (JSM-7500F, JEOL. Tokyo, Japan) at 5 kV. Observation areas were randomly selected at a magnification of 25×. Further observations were conducted at magnifications ranging from 1000× to 30,000×.

### 2.3. Evaluation of Physical Properties and Retention of Regenerated NBs

The material containing NBs was dissolved in the same 3.6 mL of distilled water as before lyophilization. The physical characteristics of the NBs were measured using nanoparticle tracking analysis, flow cytometric analysis, and resonance, as described previously [36,37,38].

#### 2.3.1. Nanoparticle Tracking Analysis

NB particle size was measured using a nanoparticle tracking analysis (NTA) device (NanoSight LM10; Malvern Instruments, Worcestershire, UK). The nanoparticle suspension was illuminated using a 638 nm wavelength red laser. Nanoparticle movement was visualized by light scattering, and Brownian motion was recorded using a CCD camera (C11440-50B; Hamamatsu Photonics K.K., Shizuoka, JP). This system automatically detects the center position of a nanoparticle and tracks its motion in a two-dimensional plane for later calculation of the average moving distance under Brownian motion. The summated image of a particle’s movement with NTA was recorded for 60 s at room temperature. The range of particle size measurements using the NTA method was adjusted from 10 to 1000 nm. The particle size was estimated from the average moving distance using the Stokes–Einstein equation. An NB suspension of 0.5 mL was injected into the sample measurement chamber of the Nanosight system with a 1.0 mL volume plastic syringe (Terumo, Tokyo, Japan). Sample image capturing and data analysis were performed using NanoSight NTA 3.4 (Dev Build 3.4.4). All measurements were performed independently for each sample. Particle size is presented as a mean and mode ± standard error of the average of 3 measurements. The size of the regenerated NBs after dissolution was compared with that of the centrifuged sample without lyophilization.

#### 2.3.2. Flow Cytometric Analysis

The proportion and number of NBs were measured using a flow cytometer (CytoFLEX; Beckman Coulter, Pasadena, CA, USA). The flow cytometer was equipped with a 405 nm (violet) laser to detect nanoparticles and was used to measure the Side Scatter (SS) from the violet laser for enhanced nanoparticle detection (Violet-SS). The Violet-SS signal-resolution limit for particle detection was 200 nm. This method was chosen because superior resolution can be obtained with SS compared to the forward scatter signal, and SS is suitable for measuring small particles (e.g., nanoscale particles). To relate the Violet-SS Area (Violet SS-A) to particle size, we calibrated the flow cytometer with beads of known size [39,40]; polystyrene standard beads (200 nm, qNano Calibration Particles, Izon Science, Christchurch, NZ and 500 nm, Archimedes Standard polystyrene beads, Malvern Instruments, Worcestershire, UK) were suspended in ultrapure water and analyzed. The acquired Violet SS-A signals of the NBs were then analyzed using CytExpert analysis software (version 2.4.0.28; Beckman Coulter, CA, USA). The gates of the Violet SSC-A values were created based on the size of each standard bead in the range of 200–500 nm to determine the size of the fabricated NBs. Using these data, the number of NBs present in each signal band was measured. The number of NB particles was diluted 10-fold before taking any measurements, and the concentration of the stock suspension was retrospectively calculated. Based on the concentration of NBs in the centrifuged solution without lyophilization, the percentage remaining after lyophilization-dissolution was calculated for the total and each size range.

#### 2.3.3. Resonance Mass Measurement

To confirm that the nanoparticles present after lyophilization and dissolution were indeed buoyant bubbles, the particle masses were measured using a Resonance Mass Measurement (RMM) system (Archimedes; Malvern Instruments Ltd., Worcestershire, UK), based on our previous report [37]. RMM was employed such that the sample solution passed through a microfluidic flow channel inside the cantilever. The particles that passed through the microfluidic flow channel were detected because of a momentary shift in the resonant frequency of the cantilever associated with the change in mass caused by the passage of a particle of differing density from the solution. The direction of the frequency shift clearly distinguishes particles with positive and negative buoyancy [41,42]. A resonator in the Archimedes Hi-Q nanosensor (Malvern Instruments Ltd., Worcestershire, UK) with internal microfluidic flow channel dimensions of 2 × 2 μm^2^ was used in our experiments. For all measurements, the limit of detection (or a threshold of 0.01 Hz) was manually selected based on the observed baseline noise in the control samples of the phosphate buffered saline (PBS) solution. The NB suspension was supplied to the Hi-Q nanosensor and measured continually for 20 min at room temperature. The buoyant mass was calculated from the transitory resonant frequency shift using Particle Lab Software version 1.9.81 (Malvern Instruments Ltd., Worcestershire, UK).

### 2.4. Ultrasound Responsiveness of Regenerated Nanobubbles

To confirm that the nanoparticles present after lyophilization and dissolution were indeed US-responsive bubbles, their size distribution was measured before and after sonication using a method based on a previous report [36]. Briefly, the NB suspension (100 μL) was placed within an acoustically transparent film based 96 multi-well cell culture polystyrene plate (lumox multiwell; Sarstedt, Nümbrecht, NRW, DE). The culture plate was fixed above the surface of the US transducer using an acoustic transmission gel (Aquasonic 100 gel; Parker Lab, Fairfield, NJ, USA). US was applied using a sonoporator (SP100; Sonidel Limited, Dublin, Ireland) with a transducer (diameter 1.6 cm), driving frequency of 1 MHz, burst rate of 100 Hz, and duty ratio of 50% and intensity of 1.0, 2.0 or 5.0 W/cm^2^ for 30 s (Figure 1A). The US irradiation method was identical to a previously described micro-scale in vitro sonoporation system using a 96 multi-well plate that included cultured cells. The diameters of the NBs were measured after sonication at various intensities (0, 1, 2, and 5 W/cm^2^) for 30 s. Changes in the number and distribution of NB particles before and after US irradiation were measured using NTA, flow cytometry (FCM), and RMM, as described above.

### 2.5. In Vitro US Image Characterization of Regenerated NBs

To evaluate the function of regenerated NBs as US contrast agents, US imaging was performed.

A ready-made flow phantom (7 cm × 8 cm × 11 cm, Model VP3; Eastek, Tokyo, Japan) (Figure 2) was set up for the US contrast-enhanced imaging experiment. The NB and control solution were injected separately through two acoustically transparent vessels (5 mm φ) placed parallelly, 2 cm distance apart. Both vessels were placed horizontally, 1.5 cm deep from the surface of the phantom and probe. The regenerated NB solution or control, diluted 3-fold with distilled water, was filled into a syringe and injected at a rate of 2 mL/min directly into the flow vessel in the same direction using an auto-injector (YSP-201; Terumo, Kyoto, Japan) through polyvinyl chloride tubes with an inner diameter of 3.1 mm (SF-ET2022L; Terumo, Tokyo, Japan) connected to both holes of the flow vessels. Acoustic evaluation of the regenerated NBs was performed using a diagnostic US imaging system (LOGIQ E9; GE Healthcare, Chicago, CHI, US) with a broad-spectrum interoperative linear array L8-18i-D probe (4–14 MHz). The US probe is fixed to the top surface of the flow phantom. B-mode images of the US flow phantom were acquired with and without the contrast mode of Coded Harmonic Angio (CHA, mechanical index 0.6) or Amplitude Modulation (AM, mechanical index 0.28).

### 2.6. In Vitro Luciferase mRNA Transfection and Evaluation of Expression

#### 2.6.1. mRNA Transfection

Confirmation of the intracellular delivery of mRNA by the US responsiveness of regenerated NBs was conducted as previously reported [36]. Oral squamous carcinoma cell line HSC-2 was purchased from JCRB (Japanese Cancer Research Bank) cell bank and cultured in Minimum Essential Medium (MEM; Nacalai Tesque, Kyoto, Japan) with 10% Fetal Bovine Serum (In Vitrogen, Tokyo, Japan). Cells were maintained at 37.0 °C in humidified air with 5% CO2. HSC-2 cells collected by trypsin–EDTA (Gibco, New York, NY, USA). They were then washed and maintained in fresh medium immediately before each sonoporation experiments. On the day before the experiment, cells were collected and centrifuged at 100× *g* for 5 min. They were seeded by 3 × 10^3^/well on transparent film based 96 multi-well cell culture polystyrene plate (lumox multiwell; Sarstedt, Nümbrecht, Germany). The cell line was free of viral pathogens with initial viability of more than 99% before use in the actual experiments. Lyophilized NB-containing material was dissolved in the same volume of Opti-MEM as the aqueous solution before lyophilization. mRNA encoding Gaussia luciferase (Gluc) was added to NBs or the control solution at a final concentration of 10 μg/mL. Each HSC-2 cells culture medium replaced with 50 μL regenerated NB medium or control solution, which included 500 ng mRNA, respectively (Figure 1B,C). US (SP100, Sonidel Limited, Dublin, Ireland) was irradiated from the transducer (diameter 1.6 cm) (SONIDEL SP100, Sonidel Limited. Dublin, Ireland) to the culture plate bottom containing HSC-2 cells, NBs, and mRNA (Figure 1D). The US condition was at the driving frequency of 1 MHz, burst rate of 100 Hz, duty ratio of 50%, and intensity of 1.0, 2.0, or 5.0 W/cm^2^ for 30 s. Following this US irradiation treatment, the suspension containing the NBs was removed. From mixing the mRNA and NBs until sonication, the process was completed within 2 min. Then, each culture well was re-filled with 100 μL of culture medium and incubated at 37 °C in a humidified, 5% CO_2_ atmosphere (Figure 1E). After 24 h, the supernatant was collected, and luciferase expression and cell viability assays were performed (Figure 1F). In this experiment, seeding of HSC2 cells and mRNA transfection on 96 multi-well plate was placed, every second row and column, in order to prevent interaction of US irradiation to each other (Figure 1G).

#### 2.6.2. Luciferase Expression Assay

In vitro luciferase activity was determined using a Spark Multimode Microplate Reader (Tecan, Männedorf, Zürich, CH, USA). Briefly, following a 24 h incubation period after cell sonication, 10 μL of culture supernatant was retrieved from each incubation well and applied to a corresponding well on a Costar 96 well white solid plate (Corning, New York, NY, USA). This was followed by the injection of 1 μg/100 μL of coelenterazine (Gold Biotechnology, Olivette, MO, USA), dissolved in 0.01% Tween 20/0.1 mM EDTA/PBS, into each well. The relative luminescence unit (RLU) values from 2 s to 12 s after injection were plotted and totaled.

#### 2.6.3. Cell Viability Assay

The number of viable HSC-2 cells was measured by using the colorimetric method with 3-(4,5-dimethylthiazol-2-yl)-5-(3-carboxymethoxyphenyl)-2-(4-sulfophenyl)-2H-tetrazolium (MTS) in a cytotoxicity assay [CellTiter 96 AQueous One Solution Cell Proliferation Assay system (Promega, Madison, WI, USA)]. After adding 20 μL of Cell Titer Solution Reagent to each well, a portion of the supernatant was removed for the luciferase assay. After 2 h of incubation, absorbance was recorded at 490 nm using a 96-well plate reader (Multiskan Go, Thermo Fisher Scientific, Waltham, MA, USA). The survival rate of the treated cells was calculated as the ratio of the number of surviving cells to the number of untreated (control) surviving cells.

### 2.7. Statistical Analysis

Data are presented as mean ± standard error of the mean (s.e.m). Data were analyzed using an unpaired t-test with Welch’s correction. Statistically significant differences between groups were analyzed using Microsoft Excel (Version 2212; Microsoft, Redmond, WA, USA). A probability value of *p* < 0.05 was considered statistically significant.

## 3. Results

### 3.1. Shells of HSA-NBs Retained with Lyophilization

The HSA solution was warmed to a maximum of 25.0 °C following 4 vibration cycles, and quickly frozen and vacuum dried (Figure A1). Lyophilized materials appeared macroscopically cotton-like, with or without bubbling (Figure 3). These structures were soft and easily torn when pinched with tweezers. Visual inspection and observation of hardness using tweezers revealed no distinguishable difference between the two.

Many dried shells of bubbles over 1 μm in diameter were detected with low magnification SEM (×1000) when observing lyophilized material made of solution including bubbles. Shell pores were observed in the majority of these MBs (Figure 4A). High-magnification SEM (×10,000) revealed a spherical material, possibly NBs (Figure 4B). Spherical HSA-NBs with pores in some of their shells were confirmed, although it was difficult to completely distinguish them from the irregularly, snaggle-shaped albumin aggregates that intermingled with them under ultra-high magnification SEM (×30,000) (Figure 4C). Only irregularly shaped albumin aggregate structures were present, without structures that appeared to be MB or NB shells at low- to very high-magnification SEM (×1000–×30,000) when observing the lyophilized materials of solutions without bubbling (Figure 4D–F).

### 3.2. NBs Are Regenerated by the Dissolution of Lyophilized Material That Includes HSA Shells

The material containing HSA-NBs was redissolved in an equal volume of distilled water before lyophilization. Particle concentrations and distributions before and after lyophilization were back-calculated from measurements obtained via FCM analysis diluted 10-fold (Figure 5A, Table A1). The concentration of HSA-NBs in the solution before vacuum lyophilization was 1.2 × 10^9^/mL. In comparison, the concentration of bubbles in the regenerated NB solution was 7.8 × 10^8^/mL. Following vacuum freeze-drying, 64.5% of the NBs were retained. NBs <200 nm in diameter were preserved at 102.0% (3.0 × 10^8^/mL to 3.1 × 10^8^/mL). NBs 200–500 nm and >200 nm were preserved only 57.3% (7.9 × 10^8^/mL to 4.5 × 10^8^/mL) and 17.7% (1.2 × 10^8^/mL to 2.2 × 10^7^/mL), respectively.

Using NTA, the concentration of NBs was found to have been decreased from 13.8 × 10^9^/mL before lyophilization to 4.8 × 10^9^/mL (34.8%) after lyophilization (Figure 5B). The average NB size was 266.7 ± 17.1 nm before vacuum freeze-drying, but it reduced to 224.8 ± 13.8 nm after vacuum freeze-drying.

Using RMM, the concentration of NBs was found to have been decreased from 11.5 × 10^7^/mL before lyophilization to 9.6 × 10^7^/mL (83.2%) after lyophilization (Figure 5C). Moreover, 100.0% of the NBs present before vacuum freeze-drying had a positive buoyancy, with an average buoyant mass of −2.6 fg. In contrast, 97.7% of the NBs present after vacuum freeze-drying had a positive buoyancy, with an average buoyant mass of −2.1 fg.

### 3.3. US Irradiation Collapses NBs Regenerated from Lyophilized Materials

The NB solution regenerated from lyophilized material was irradiated with US. US irradiation intensities of 1, 2, and 5 W/cm^2^ reduced the concentration of NBs from 6.3 × 10^8^/mL to 2.1 × 10^8^/mL (33.3%), 1.6 × 10^8^/mL (24.7%), and 9.9 × 10^7^/mL (15.7%), respectively, according to FCM analysis (Figure 6A, Table A2). The concentrations decreased from 2.0 × 10^8^/mL to 7.1 × 10^7^/mL (34.7%) for NBs < 200 nm in diameter, from 4.0 × 10^8^/mL to 2.5 × 10^7^/mL (6.2%) for NBs 200–500 nm, and from 2.2 × 10^7^/mL to 2.5 × 10^6^/mL (11.3%) for NBs >500 nm, using an intensity of 5 W/cm^2^.

US irradiation intensities of 1, 2, and 5 W/cm^2^ reduced the concentration of NBs from 35.3 × 10^8^/mL to 18.1 × 10^8^/mL (51.3%), 7.1 × 10^8^/mL (20.0%), and 5.2 × 10^8^/mL (14.8%) respectively, according to NTA (Figure 6B). The average NB size decreased from 266.1 ± 6.7 nm to 189.3 ± 11.1 nm, 192.5 ± 3.6 nm, and 166.8 ± 2.2 nm at intensities of 1, 2, and 5 W/cm^2^, respectively.

The concentration of NBs decreased from 95.7 × 10^6^/mL to 26.0 × 10^6^/mL (27.2%), 4.4 × 10^6^/mL (4.6%), and 5.49 × 10^6^/mL (5.7%) at US intensities of 1, 2, and 5 W/cm^2^, respectively, according to RMM. The average buoyant mass of the NBs was −2.1 fg before sonication and −1.9, −0.3, and −1.25 fg after sonication, respectively. Particles with positive buoyancy accounted for 97.7% of the total number of particles before sonication, and 98.4, 76.5, and 100.0% after sonication at US intensities of 1, 2, and 5 W/cm^2^ (Figure 6C).

### 3.4. NBs Regenerated after Lyophilization Are Echogenic

Diluted solutions with and without NBs regenerated from the lyophilized material were refluxed into a flow phantom vessel to observe their echogenicity. Changes in the echo brightness in the vessel due to solution perfusion, with or without NBs, could not be detected using B-mode imaging alone (Figure 7A). An increase in brightness was detected only in the perfusion of solutions containing NBs using CHA (Figure 7B). No increase in brightness was observed with or without NBs in the solution using AM (Figure 7C).

### 3.5. NBs Regenerated from Lyophilized Materials Act as Cavitation Nuclei in mRNA Sonoporation

mRNA transfection efficiency increased stepwise with and without NBs, as observed with the stepwise increase in the acoustic intensity of US at 1, 2, and 5 W/cm^2^ in vitro sonoporation (Figure 8A). This increase in the mRNA transfection efficiency after sonication was greater in the presence of NBs. RLU values in the conditions with and without HSA-NBs without US irradiation were 5.9 ± 1.0 (×10^5^) and 1.6 ± 0.3 (×10^5^), respectively (*p* = 0.02987). The RLU value in the condition with NBs and US irradiation at the maximum acoustic intensity of 5 W/cm^2^ reached 2.6 ± 0.1 (×10^7^). This value was 7.5 times higher than that of the transfection efficiency of mRNA at the same US intensity using a solution without NBs (*p* = 0.0002). Cell viability after US irradiation at 5 W/cm^2^ was 84.9% and 90.7% with and without NBs, respectively (*p* = 0.6255) (Figure 8B).

## 4. Discussion

The combination of US irradiation and bubble reagents is an important tool in medicine. This technology is widely used in clinical practice as an US contrast agent to diagnose various diseases, including liver and breast cancers [4]. It is also considered a promising technology for many drug and gene delivery systems [43]. However, several limitations remain, including the difficulty of storing bubble reagents for long periods. Therefore, the preservation and purification of bubble reagents must be improved to expand their therapeutic application in clinical settings. This study revealed that albumin-based NBs can be preserved, and their size can be decreased by lyophilization.

One of the characteristics of the bubble reagents used in medical applications and research is that they are stabilized by shells composed of various compounds. MB and NB shells can be composed of phospholipids [44,45], polymers [46,47], or proteins such as HSA [48,49,50,51]. HSA, which consists of three flexible spheres (domains I, II, and III) and two binding sites that provide pockets for small molecules, mainly aromatic dyes (binding site 1) and lipophilic carboxylic acid derivatives (binding site 2) [52,53], is an abundant protein in the human body and is thus highly biocompatible. Therefore, HSA is a useful carrier for drug delivery to various organs and drug retention [54,55]. As such, the Food and Drug Administration has approved several HSA-based formulations in the past two decades [56].

Protein-shell bubbles are a compromise between the vibrational profile of soft lipid-shell bubbles and the drug-loading capacity of hard polymer-shell bubbles with moderate properties [57]. The thickness of protein-shell bubbles is known to be approximately 15–20 nm, between that of lipid and polymer shells [57,58]. As such, HSA-based bubble reagents can be loaded with various low-molecular-weight compounds, including anticancer drugs, such as doxorubicin [59] and cisplatin [60], and proteins, such as IgG antibodies [61]. Albumin-shell bubbles can also carry pDNA [62,63] and viruses containing integrated genes [64]. Furthermore, HSA-based bubbles can be PEGylated to improve stability [50]. In our previous study, we developed albumin-based NBs and investigated their characteristics. Carrier-free pDNA [38] and mRNA [36] delivery without any artificial compounds has been achieved using NBs whose shells are composed of albumin, a complete biomolecule.

Certain proteins, such as HSA, exhibit foaming properties. Conformational changes in protein molecules such as HSA are induced in water, placing them at the gas–liquid interface. Proteins that reach the gas–liquid interface of the bubble expose the hydrophobic region of the molecule to the gas phase and are replaced by water molecules in a higher-energy state [65]. Denaturation of albumin molecules by superheating increases the efficiency of their encapsulation. Previous biochemical analyses have suggested that HSA-based MB shells are formed by layers of native and denatured monomolecules in multiple orientations [66]. Alternatively, the superoxide produced at high temperatures by US forms protein-to-protein crosslinks of cysteine residues, stabilizing the HSA-based MB shell [67]. Consequently, the bubbles are stabilized by a protein shell formed on the surface. Our experimental results showed that HSA forms and stabilizes MB and NB shells, as seen by the SEM observation of lyophilized materials. Denaturation of HSA may have been caused by vibration-induced heating in our method, which may have contributed to the stabilization of the bubble shell.

The first marketed MB reagents to be used as US contrast agents worldwide were Albunex^®^ [58] and, subsequently, Optison^®^ [68]. The shells of these bubble reagents were composed of denatured HSA and were stable for at least 2 years in refrigerated storage. However, these discontinued formulations were never commercialized lyophilized products. A few reports have revealed a reduction in bubble size by lyophilization and improved efficiency through the freeze-dried preservation of albumin-shell MBs by addition of sugars such as dextrose and fructose [35]. By the mechanisms of glassy state theory [69,70], water substitution theory [71], and partition effect B [72], amorphous matrices composed of sugars stabilize proteins in freeze-drying. Sugars may be able to form stabilizing interactions with the protein during drying, thereby maintaining it in its native conformation and reducing both local and global mobility during storage [73]. These mechanisms may help retain proteins such as HSA, in the bubble shell even during lyophilization. Conversely, the collapsed larger MB shells observed by SEM of the lyophilized material in this study reveal the difficulty of maintaining the shell conformation of HSA molecules without a stabilizing agent. In contrast, it may be easier for NBs, which have a much smaller surface area than MBs, to maintain the conformation of albumin molecules in their shells. Thus, the size and concentration of albumin-shell NBs in this study seem to have decreased after lyophilization, as observed with FCM analysis and NTA. The collapse of bubbles with a large surface area, and the persistence of small bubbles can be explained with the reduction in bubble size of MBs by freeze-drying a per the previous study [35]. In previous reports, HSA-NBs showed a wider distribution than lipid-shell NBs [36,37,38,51]. Nanoparticles in the range of 100–200 nm are optimal for achieving the EPR effect in solid tumors while escaping the liver and spleen filtration traps [74]. HSA-NBs with diameter greater than 200 nm may not be able to adequately pass through the pores of capillaries, tissues, and blood vessels in vivo. In this study, lyophilization tended to compromise bubbles larger than 200 nm in diameter, especially those larger than 500 nm, whereas NBs smaller than 200 nm in diameter remained nearly intact. The miniaturization of NB particle size may be useful in controlling their retention and distribution in vivo.

The shared properties between regenerated nanoparticles and HSA-NBs have already been clarified. Regenerated NBs are positively buoyant, as seen in our previous report [51]. This is evidence that the nanoparticles regenerated after lyophilization were not dust, such as albumin aggregates, but bubbles containing gas. Regenerated NBs after lyophilization showed ultrasound responsiveness consistent with our previous reports [36,37,38]. Specifically, ultrasound responsiveness refers to the disappearance of particles larger than 200 nm, echogenicity, and gene delivery ability in response to ultrasound irradiation. As in our previous report, regenerated HSA-based NBs also functioned as US contrast agents and cavitation nuclei for the intracellular delivery of genes [36,37,38]. These results also suggest that regenerated nanoparticles are bubbles fitted with the Rayleigh−Plesset equation [6] and the Minnaert resonance phenomenon [5]. Notably, HSA-based NBs were detected only by CHA and not by AM. Previous studies have reported that NBs are non-echogenic in basic-mode contrast-enhanced US imaging [75]. AM imaging is a method of extracting nonlinear harmonic signals in the fundamental band scattered from contrast agent bubbles by transmitting and receiving two pulsed waves of different amplitudes [2]. In contrast, CHA imaging suppresses the tissue-derived signal from the second-harmonic signal obtained by phase inversion and extracts only the contrast bubble-derived signal based on proprietary digitally coded US technology of General Electric Company (GE) [76,77]. Our experimental results reveal that the selective extraction of the second harmonic allows HSA-based NBs with low echogenicity to function as excellent contrast agents. These results indicate that HSA-NBs were preserved in lyophilized material and regenerated by dissolution.

Previously, we established carrier-free mRNA sonoporation in vitro [36], and in this study, we confirmed that this method also works with NBs that have undergone freeze-drying. At present, however, mRNA delivery using this method remains restricted to in vitro experiments. Interestingly, we observed the slight increase in mRNA delivery efficiency in the presence of NBs alone, without ultrasound. The reason that nanobubbles alone resulted in increase of gene transfer is unclear. However, previous studies have shown that in sonoporation, besides direct perforation of the cell membrane, clathrin-mediated endocytosis [78] and caveolae endocytosis [79] can occur. Our results suggest that the gas itself contained in NBs may have facilitate endocytosis slightly without ultrasound.

To deliver mRNA via blood flow to deep organs in vivo using sonoporation, loading onto a carrier is required to avoid RNase attack [80,81,82]. In contrast, directly reachable organs such as skin and muscle may be optimal targets for carrier-free mRNA delivery by combining bubble reagents and US. In contrast, PEG modifications made to the lipid nanoparticle (LNP) carriers of existing mRNA vaccines have been reported to cause side effects such as anaphylactic shock [83]. A possible solution could be generation of RNase-free HSA-NBs, freeze-dried together with carrier-free mRNA. This would ensure stability of the mRNA without the need for existing carriers. Injection of the subsequently regenerated RNase-free NBs and mRNA in vivo, synchronously with ultrasound irradiation, may be a reasonably efficient strategy for delivery. It is safe to say that our present lyophilization studies opens new grounds for a potential application for HSA-NBs and mRNA delivery in vivo for mRNA vaccination into skin or muscle.

## 5. Conclusions

In conclusion, this study demonstrated that freeze-drying and redissolving operations can maintain albumin-shell NBs at a relatively high concentration. Therefore, freeze-drying is a suitable method for completely removing MBs from bubble solutions and refining NBs. Control of NB particle size by lyophilization may contribute to future control of retention and distribution in vivo but is not yet sufficiently established. HSA-NBs have the potential to not only serve as a safe US contrast agent but also as a carrier-free method for mRNA vaccine delivery. However, further research is needed to increase the storage and miniaturization efficiency of this method and to add drug and gene loading capacities.

## Figures and Tables

**Figure 1 nanomaterials-13-01283-f001:**
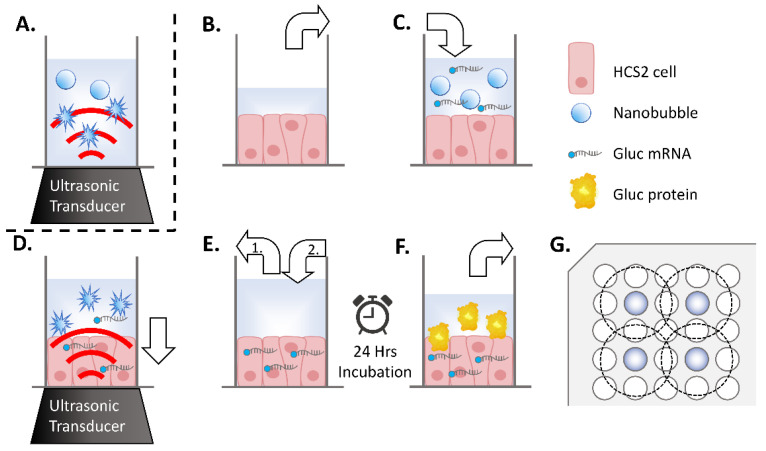
Schematic representation of sonication treatment of nanobubbles (NBs) or sonoporation in 96-well plates. (**A**) Sonication treatment of NBs by ultrasonic irradiation to medium. (**B**–**F**) Method of sonoporation. (**B**) The incubation medium is removed from the well (white arrow) of a 96 multi-well plate seeded with human squamous carcinoma (HSC-2) cells. (**C**) Each well is filled with medium containing NBs (white arrow). (**D**) Transfection by ultrasonic irradiation. mRNA is transferred into the cytoplasm (white arrow). (**E**-1) Aspiration of sonicated medium and (**E**-2) addition of new incubation medium. (**F**) After a 24 h incubation period, the supernatant was collected for a reporter assay. (**G**) Arrangement of wells of seeded cells (indicated with color) and ultrasonic irradiation area (inside of dashed circle) on a 96 multi-well plate.

**Figure 2 nanomaterials-13-01283-f002:**
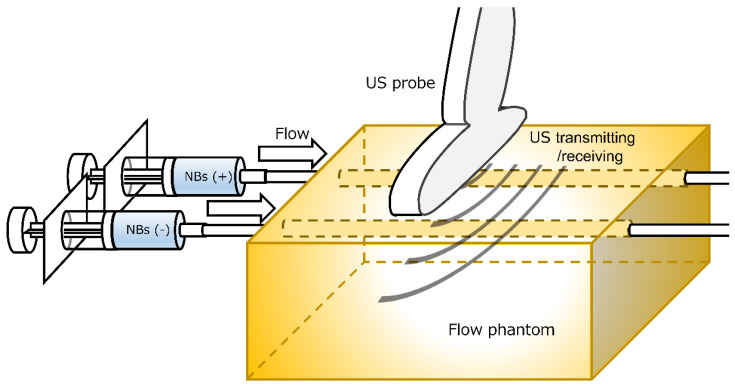
Flow phantom and measurement set up to characterize regenerated NBs with ultrasound. NBs (+): solution with nanobubbles; NB (−): solution without nanobubbles.

**Figure 3 nanomaterials-13-01283-f003:**
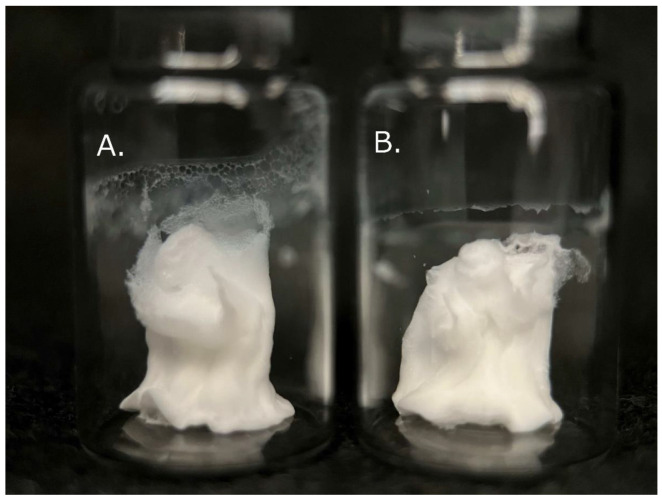
Appearance of lyophilized human serum albumin (HSA)-based solutions with (**A**) and without (**B**) bubbling.

**Figure 4 nanomaterials-13-01283-f004:**
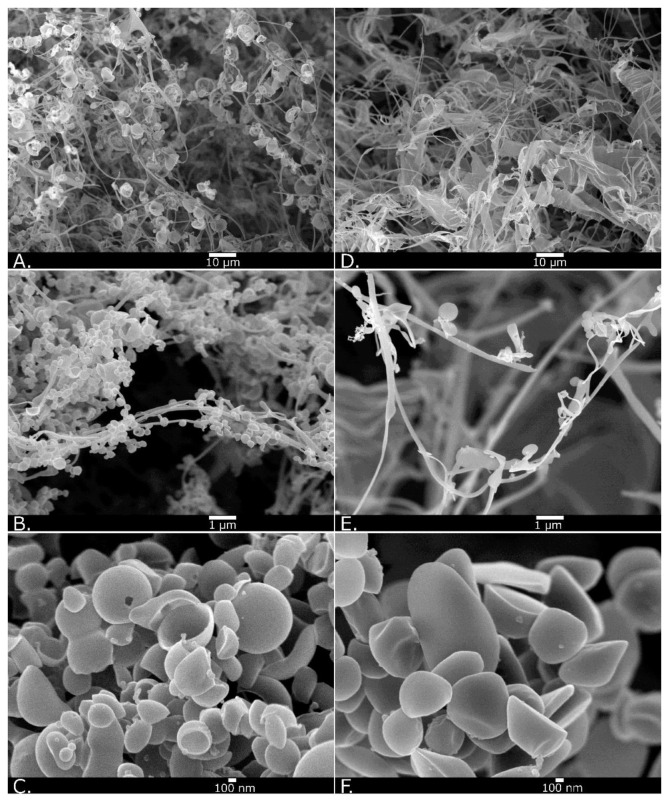
Scanning electron microscopy images of lyophilized material in HSA solution with and without bubbling. Low magnification (**A**; ×1000), high magnification (**B**; ×10,000), and ultra-high magnification (**C**; ×30,000) of bubbled material. Low magnification (**D**; ×1000), high magnification (**E**; ×10,000), and ultra-high magnification (**F**; ×30,000) of material without bubbling.

**Figure 5 nanomaterials-13-01283-f005:**
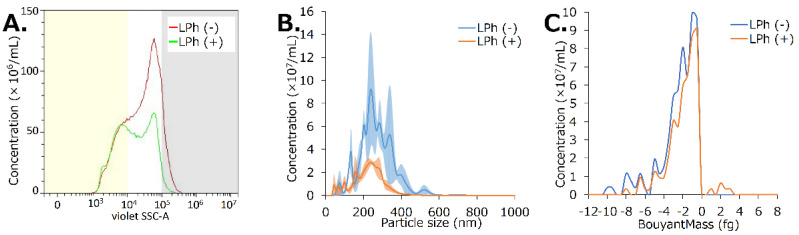
Characteristic changes of bubbles in solution before and after lyophilization. (**A**) Comparison of bubble concentration and size distribution using flow cytometry (FCM) analysis. (**B**) Comparison of bubble concentration and particle size using nanoparticle tracking analysis (NTA). (**C**) Comparison of bubble concentration and buoyant mass using Resonance Mass Measurement (RMM) analysis. LPh (−): solution before lyophilization; LPh (+): solution after lyophilization.

**Figure 6 nanomaterials-13-01283-f006:**
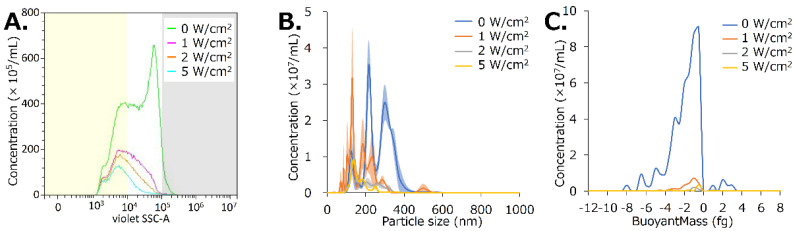
Characteristic changes of bubbles in solution after lyophilization by sonication. (**A**) Comparison of bubble concentration and size distribution using FCM analysis. (**B**) Comparison of bubble concentration and particle size using NTA. (**C**) Comparison of bubble concentration and buoyant mass using RMM analysis.

**Figure 7 nanomaterials-13-01283-f007:**
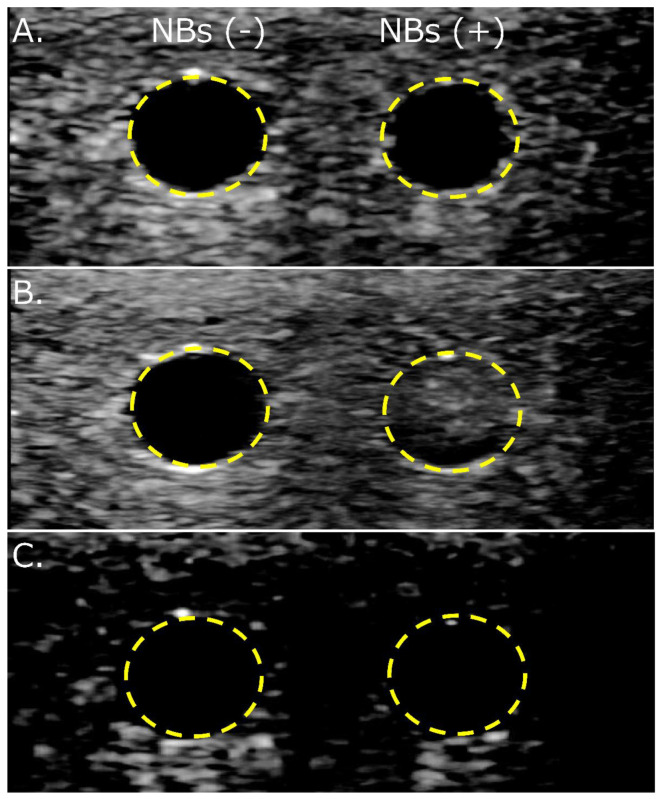
Detection of changes in echo brightness in solution with and without nanobubbles regenerated from lyophilized material using flow phantom vessels. (**A**) B-mode only. (**B**) B-mode with Coded Harmonic Angio (CHA). (**C**) B-mode with Amplitude Modulation (AM). NBs (−): solution without nanobubbles, NBs (+): solution with nanobubbles.

**Figure 8 nanomaterials-13-01283-f008:**
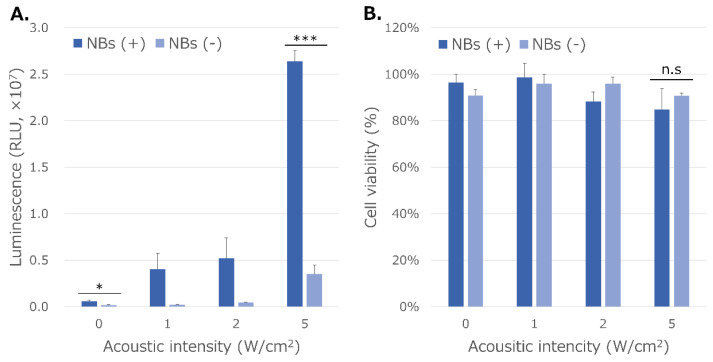
mRNA transfection efficiency (**A**) and cell viability (**B**) by sonoporation using NBs regenerated from lyophilized material. NBs (+): solution with nanobubbles, NBs (−): solution without nanobubbles. (* *p* < 0.05; *** *p* < 0.001; n.s: not significant) (*N* = 3).

## Data Availability

Not applicable.

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
