# Peer review of "Efficient mRNA Delivery with Lyophilized Human Serum Albumin-Based Nanobubbles"

_nanomaterials, 2023, doi:10.3390/nano13071283_

Round 1

Reviewer 1 Report

The authors presented the paper "Efficient mRNA Delivery with Lyophilized Human Serum Albumin-based Nanobubbles"

1) The reference list should be improved. Many more 2-3 years review papers should be cited in the Introduction section to show the progress in the area. I highly recommend not to use the references older 10 years for this section if it is possible.

2) Dear author, according to the literature data the size of nanoconstructions is optimal in the range from 20 nm to 200 nm (better 150 nm) (for in vivo applications). Too-high-sized nanoparticles (> 200 nm) do not pass through capillaries and tissue and vessel pores. Please, mention these limitations in the Conclusion section. Moreover, please explain how this high-sized system will work as a drug delivery system. Please, enlarge the discussion about it.

However, I recommend to avoid using "nano"-bubles for so high sized system.

NBs < 200 nm in diameter were preserved at 102.0% (3.0×108 /mL to 303 3.1×108 /mL). NBs 200–500 nm and >200 nm were only 57.3% (7.9×108 /mL to 4.5×108 /mL) 304 and 17.7% (1.2×108 /mL to 2.2×107 /mL) preserved, respectively.

Please, correct that you have, I think, 10.2% instead of 102.0% lower than 200 nm.

Moreover, what the size will be in solution? Have you used dynamic light scattering?

3) Albumin has RNAase activity. In this way have you studied if the RNA is stable in your solutions?

4) How you can explain that mRNA transfection efficiency increased in the presence of your albumin nanobubles. Please, add some discussion of this phemomenon in subsection 3.5 and discussion.

5) I see that your approach work on the cell lines. However, how it will be done on in vivo system. Can you add some discussion, maybe a picture, etc. In other way, it is a bit confused. Moreover, mRNA is not stable and can't be used alone without any protection for in vivo studies. That is why its transfection without the protection is an ideal in vitro experiment.

Author Response

Response 1)

As the reviewer pointed out, the 1990s-2000s reviews cited were indeed too old. The references to be cited have changed along with the advances in bubble agents and ultrasound.

Specifically, the references numbered [1], [2], [3], [7], [16], and [27] have been updated. In contrast, the important original papers that were breakthroughs in these studies were left intact. (Refs.[5],[6],[8],[9], [10], [15], [21] and [23])

Response 2)

  1. I agree with the reviewer's point. Not all nanobubbles less than 1 µm in diameter with a wide distribution would cause the EPR effect. In fact, few studies have shown that nanobubbles come with the same reliable EPR effect as other nano-sized DDS. Several corrections were made to the manuscript. The text in the introduction has been rewritten as follows (in line 72-74):

" EPR effect is expected to occur only in NBs that are particularly small, specifically, less than 200 nm in diameter."

The following text and references were added to the Discussion (in line 455-462).

“In previous reports, HSA-NBs showed a wider distribution than lipid-shell NBs. Nanoparticles in the range of 100-200 nm are optimal for achieving the EPR effect in solid tumors while escaping the liver and spleen filtration traps. HSA-NBs with diameter greater than 200 nm may not be able to adequately pass through the pores of capillaries, tissues, and blood vessels in vivo. In this study, lyophilization tended to compromise bubbles larger than 200 nm in diameter, especially those larger than 500 nm, whereas NBs smaller than 200 nm in diameter remained nearly intact. The downsizing of NB particle size may be useful in controlling their retention and distribution in vivo."

  1. I am sorry that the description is not clear. The percentage of NB less than 200 nm in diameter remaining is "102.0% (3.0 x 10^8/mL to 3.1 x 10^8/mL)" as measured by using FCM. Notably, freeze-drying preserves NBs less than 200 nm in diameter almost completely. This can be seen from the overlapping lines before (Red) and after (Green) freeze-drying in Fig. 5A below 200 nm diameter (yellow region).This suggests that freeze-drying may be a useful method in terms of nanobubble size miniaturization. Downsized nanobubbles may be more likely to exhibit EPR and other effects. This point is discussed in Line 447-456 above.

  1. These measurements are in a solution that was redissolved in an equal volume (3.6 ml/bottle) of deionized water as before lyophilization. This is noted in the material and methods in line 125 to 128. As the reviewer point out, it is very important to measure the size of nanobubbles by using multiple methods. In a previous study, we measured and compared HSA-NBs to NTA and DLS.( Lafond M, et al. Sci Rep. 2018 May 10;8(1):7472). In this study, instead of DLS, three different methods were used: nanoparticle tracking analysis, flow cytometry, and resonance mass measurement. These results showed consistent trends in particle concentration and size changes due to lyophilization.

Response 3).

As you pointed out, degradation by RNase is a major barrier in mRNA transfection. A large amount of RNase must have been present in the human serum albumin solution and culture medium used in this study. Although not confirmed, one would probably expect that if mRNA was added to the medium for an extended period, it would be degraded by RNase. The key point in this experiment is that the high concentration of mRNA and nanobubbles are mixed and delivered into the cell via sonoporation within a short time (2 min), when mRNA degradation does not occur. It is important that the mixing and US irradiation take place immediately. A detailed procedure has been added to line 253 to 253 of the materials and methods.

" From mixing the mRNA and NBs until sonication, the process was completed within 2 min.”

Response 4)

We agree with the reviewer's point. Indeed, in Figure 8A, without ultrasound irradiation (0 w/cm2), the mRNA delivery efficiency is slightly improved in the presence of HSA-NBs. This is certainly an interesting point: both the NBs solution and the control solution have the same composition except for the enclosed gas. This result suggests that the gas itself may slightly promote endocytosis.

The following text has been added to Result section 3.5. (in line 371-373)

“RLU values in the conditions with and without HSA-NBs without US irradiation were 5.9 ± 1.0 (×10^5) and 1.6 ± 0.3 (×10^5), respectively (p = 0.02987)”.

The results of the statistical analysis without irradiation have been added to Figure 8A.

The following text was added to the Discussion. (in line 479-493)

“Previously, we established carrier-free mRNA sonoporation in vitro, and in this study, we confirmed that this method also works with NBs that have undergone freeze-drying. At present, however, mRNA delivery using this method remains restricted to in vitro experiments. Interestingly, we observed the slight increase in mRNA delivery efficiency in the presence of NBs alone, without ultrasound. The reason that nanobubbles alone resulted in increase of gene transfer is unclear. However, previous studies have shown that in sonoporation, besides direct perforation of the cell membrane, clathrin-mediated endocytosis (Ultrasound Med Biol. 2015 Jul;41(7):1913-26.) and caveolae endocytosis (Phys Biol. 2015 Nov 24;12(6):066007) can occur. Our results suggest that the gas itself contained in NBs may have facilitate endocytosis slightly without ultrasound.”

To detail the conditions of mRNA transfection, a description of the cells preparation, well placement, and other conditions was added (in line 233-243, 246-247, 257-259).

Response 5)

I agree with the reviewer's point. It is very important to discuss the potential future benefits of nanobubble freeze-dried and carrier-free mRNA delivery technologies. At present, the demonstrated scope of these technologies is limited to in vitro studies. However, there is potential for future applications in the administration of mRNA drugs, e.g., vaccines, to directly deliverable tissues such as the skin and skeletal muscle.

The following text was added to the discussion in line 494-505.

“To deliver mRNA via blood flow to deep organs in vivo using sonoporation, loading onto a carrier is required to avoid RNase attack. In contrast, directly reachable organs such as skin and muscle may be optimal targets for carrier-free mRNA delivery by combining bubble reagents and US. In contrast, PEG modifications made to the lipid nanoparticle (LNP) carriers of existing mRNA vaccines have been reported to cause side effects such as anaphylactic shock. A possible solution could be generation of RNase-free HSA-NBs, freeze-dried together with carrier-free mRNA. This would ensure stability of the mRNA without the need for existing carriers. Injec-tion of the subsequently regenerated RNase-free NBs and mRNA in vivo, synchro-nously with ultrasound irradiation, may be a reasonably efficient strategy for delivery. It is safe to say that our present lyophilization studies opens new grounds for a potential application for HSA-NBs and mRNA delivery in vivo for mRNA vaccination into skin or muscle.”

Reviewer 2 Report

In the submitted manuscript, the authors reported the successful vacuum lyophilization of NBs with HSA shells. This is an interesting work, however, there are still some issues that should be addressed. The manuscript could be published in Nanomaterials after revised the issues listed as below.

1.      The abstract is too long-winded and needs to be refined.

2.      The font in Figure 5 and Figure 6 is too small to see clearly, authors should make the font larger.

3.      The format of the references is not uniform and should be revised.

4.      The figure below Figure 8 is again Figure 1.

Author Response

Response 1).

I agree with the reviewer's remarks. The abstract has been shortened and rewritten. Detailed figures were stripped out, and an abstract was provided for the main subject of this manuscript—the lyophilization technique of albumin-based nanobubbles.

Response 2)

As the reviewer pointed out, the text in Figures 5&6 was too small. They were replaced with figures of appropriate font size.

Response 3)

As the reviewer pointed out, the journal abbreviations in the references were not properly spelled out. These forms have been corrected.

In addition to these, several grammatical errors in the manuscript were corrected by English proofreading.

Response 4)

I apologize for the lack of clarity. After Figure 8, Figure A1 is attached as an Appendix. This Figure A1 shows the temperature change of the solution that may have affected the nanobubble formation mechanism.